# Chinese Propolis Suppressed Pancreatic Cancer Panc-1 Cells Proliferation and Migration via Hippo-YAP Pathway

**DOI:** 10.3390/molecules26092803

**Published:** 2021-05-10

**Authors:** Lingchen Tao, Xi Chen, Yufei Zheng, Yuqi Wu, Xiasen Jiang, Mengmeng You, Shanshan Li, Fuliang Hu

**Affiliations:** College of Animal Sciences, Zhejiang University, Hangzhou 310058, China; 11817025@zju.edu.cn (L.T.); 21717031@zju.edu.cn (X.C.); iriszheng92@gmail.com (Y.Z.); 11317024@zju.edu.cn (Y.W.); jxsen@zju.edu.cn (X.J.); mmyou@zju.edu.cn (M.Y.); lishanshan@zju.edu.cn (S.L.)

**Keywords:** Chinese propolis, pancreatic cancer, apoptosis, cell cycle arrest, migration, Hippo-YAP pathway

## Abstract

Pancreatic cancer is one of the most malignant cancers with high mortality. Therefore, it is of great urgency to develop new agents that could improve the prognosis of Pancreatic cancer patients. Chinese propolis (CP), a flavonoid-rich beehive product, has been reported to have an anticancer effect. In this study, we applied CP to the human Pancreatic cancer cell line Panc-1 to verify its impact on tumor development. CP induced apoptosis in Panc-1 cells from 12.5 µg/mL in a time- and dose-dependent manner with an IC_50_ value of approximately 50 µg/mL. Apoptosis rate induced by CP was examined by Annexing FITC/PI assay. We found that 48 h treatment with 50 µg/mL CP resulted in 34.25 ± 3.81% apoptotic cells, as compared to 9.13 ± 1.76% in the control group. We further discovered that the Panc-1 cells tended to be arrested at G2/M phase after CP treatment, which is considered to contribute to the anti-proliferation effect of CP. Furthermore, our results demonstrated that CP suppressed Panc-1 cell migration by regulating epithelial–mesenchymal transition (EMT). Interestingly, the Hippo pathway was activated in Panc-1 cells after CP treatment, serving as a mechanism for the anti-pancreatic cancer effect of CP. These findings provide a possibility of beehive products as an alternative treatment for pancreatic cancer.

## 1. Introduction

Pancreatic cancer, a gastrointestinal malignant tumor, is one of the most aggressive cancers with an overall 5-year survival rate lower than 10% [1]. GLOBOCAN 2018 report estimates that there will be 484,486 new cases of pancreatic cancer and 456,280 deaths caused by the disease in 2020 [2]. The occurrence and development of pancreatic cancer is a complex process, and smoking [3], alcohol [4], obesity [5], diabetes [6], chronic pancreatitis [7] are reported to contribute to pancreatic cancer, but its pathogenesis is still unclear. Although the detection and treatment of pancreatic cancer have advanced greatly in recent years, the incidence and mortality rate of pancreatic cancer has kept rising for the last decade [8]. The high mortality rate of pancreatic cancer is mainly ascribed to the difficulty in the early-stage diagnosis and the lack of efficient treatment. Lacking specific symptoms making it particularly difficult to diagnose pancreatic cancer at an early stage [9], resulting in a situation that most patients are diagnosed at an advanced stage with high metastases [10] and diffuse malignant peritoneal mesothelioma [11]. Currently, surgical resection is considered the only curative treatment for pancreatic cancer, but only 20% of patients are suitable for it [12]. Even those patients who have undergone resection have a high risk of recurrence [13]. Adjuvant methods like chemotherapy and radiotherapy are also widely used in pancreatic cancer treatment, but there is a large proportion of pancreatic cancer patients that develop drug resistance after treatment, leading to a poor prognosis [14]. Therefore, it is urgent to discover novel therapeutic methods to treat pancreatic cancer.

The Hippo-YAP signaling pathway is responsible for adjusting the organ size, maintaining the dynamic balance between cell proliferation and apoptosis, as well as suppressing tumor development [15]. When the growth-inhibitory signal is transmitted to the cell membranes, the Hippo signaling pathway is activated, eventually leading to the phosphorylation of downstream effector YAP. YAP is an oncogene that can promote excessive cell proliferation by entering into nuclear and functioning as a transcription factor. Phosphorylated YAP (p-YAP) will bind to 14-3-3 protein in the cytoplasm and be degraded, thus regulating cell proliferation [16,17]. The inactivation of the Hippo pathway and overexpression of YAP have been observed in different malignancies, including liver cancer [18], colorectal cancer [19], and gastric cancer [20], and are associated with poor prognosis [21,22,23]. Growing Evidences showed that the Hippo pathway plays a critical role in the development of pancreatic cancer [13,22]. Increasing nuclear localization of YAP has been observed in pancreatic malignancy [21]. Deletion of YAP in the pancreas effectively suppressed the progression of pancreatic cancer in a Kras mutant mouse model [24]. Moreover, YAP overexpression can promote EMT by activating the AKT cascade, eventually enhancing chemoresistance in pancreatic cancer [22]. Taken together, these studies indicate that the Hippo-YAP pathway has the potential to be a new target for pancreatic cancer treatment. YAP, as an oncogene, can promote excessive cell proliferation and resist apoptosis. After inactivation of YAP phosphorylation, the expression of downstream target genes that promote cell proliferation decreases, thereby regulating the balance of cell proliferation and apoptosis and maintaining the normal volume of organs.

Propolis is a natural resinous substance that honey bees (*Apis mellifera* L.) collect from plant buds and is exuded in combination with bee’s wax and salivary glands secretions [23]. Bees use propolis to build their hives and protect their colony from disease [24]. Propolis obtains various bioactive activities, such as antioxidant [25], anti-inflammatory [26], immunomodulatory [27], and antitumor effects [28]. The bioactive activities of propolis are dependent on its chemical composition, which is mainly affected by its botanical origin. Chinese propolis (CP) is generally of the poplar type whose main chemical components are flavonoids and phenolic compounds [29,30,31]. CP has a long history of being used in Chinese folk medicine. Increasing research has reported its anticancer effect on different cancer types, including breast cancer [32], liver cancer [28], and melanoma [33]. As a natural product, CP may provide a new method for cancer treatment. It was reported that propolis component caffeic acid phenethyl ester (CAPE) could inhibit the orthotopic growth and EMT of pancreatic cancer Panc-1 cells accompanied by the downregulation of vimentin and Twist 2 expression [34]. Other research has demonstrated a novel methodology in hyperthermia (HT) therapy called thermal cycle (TC)-HT. The synergistic anticancer effect of TC-HT and propolis was shown in pancreatic cancer cells [35]. In this study, we administered CP to Panc-1 cells and found that CP could suppress pancreatic cancer cell viability by inducing apoptosis, promoting G2/M cell cycle arrest, and inhibiting migration in Panc-1 cells by regulating EMT. Furthermore, we uncovered the mechanism underlying the anti-pancreatic cancer effect of CP, which is by activating the Hippo-YAP pathway in Panc-1 cells.

## 2. Results

### 2.1. Chemical Composition of CP Identified by High-Performance Liquid Chromatography (HPLC)

The main components in ethanol extract of CP were identified by HPLC analysis (Figure 1). Result (Table 1) showed that chrysin (42.23 mg/g), galangin (38.94 mg/g), pinocembrin (37.62 mg/g), 3-O-acetylpinobanksin (35.89 mg/g), pinobanksin (30.21 mg/g), and CAPE (15.66 mg/g) were the six most abundant components in ethanol extracts of CP. A previous study showed propolis component chrysin and galangin had a cytotoxic effect on pancreatic cells [36]. Furthermore, Duan et al. found that CAPE, which is the main constituent of Chinese propolis, suppressed the growth of pancreatic cancer in vitro [37]. Thus, we assumed that CP might exert an inhibitory effect on pancreatic cells.

### 2.2. CP Exerted Cytotoxic Effect on Panc-1 Cells

To investigate the cytotoxic effect of CP, we evaluated the viability of Panc-1 cells after treated with varying concentrations of CP (6.25, 12.5, 25, 50, 100, and 200 μg/mL) using the cell counting-8 (cck-8) assay. We found that CP exerted a significant cytotoxic effect on human pancreatic Panc-1 cells from 12.5 μg/mL in a time- and dose-dependent manner with an IC_50_ value of approximately 50 μg/mL for 48 h treatment (Figure 2A), while it did not show an inhibitory effect on HEK-293 cells (human embryonic kidney cell line) until the concentration reached 100 μg/mL (Figure 2B). According to these results, we treated Panc-1 cells with CP at concentrations of 12.5, 25, and 50 μg/mL for 48 h in the following experiments.

### 2.3. CP Induced Apoptosis in Panc-1 Cells via Intrinsic Pathway

We also examined the apoptosis rate induced by CP in Panc-1 cells using the Annexin V-FITC/PI assay. After treating cells with 12.5, 25, and 50 μg/mL CP for 48 h, the apoptotic cells significantly increased from 9.13% ± 1.76% to 34.25% ± 3.81% (* *p* < 0.01), as the concentration up to 50 μg/mL (Figure 3A).

Moreover, we verified the apoptotic effect of CP on Panc-1 cells by examining the apoptosis-related proteins. Results showed that pro caspase-3, pro caspase-8, pro caspase-9, cleaved caspase-3, cleaved caspase-8, cleaved caspase-9, and cleaved PARP were upregulated, while PARP1 full length was decreased after CP treatment for 48h (Figure 3B) in a dose-dependent manner. We further investigate the expression of mitochondrial proteins and found that the expression of Bcl-2 declined while the expression of Bax was increased, indicating mitochondrial dysfunction after CP treatment. Collectively, CP induced apoptosis in Panc-1 cells by activating the caspase cascade via the intrinsic pathway.

### 2.4. CP Induced G2/M Cell Cycle Arrest in Panc-1 Cells by Interfering with Cell Cycle Checkpoints

Studies have revealed that cell cycle arrest is an important mechanism during propolis-induced apoptosis [38,39,40]. To investigate the cell cycle distribution after CP treatment, we examined the cells with different concentrations of CP treatment for 48 h by using flow cytometry. Our data showed that Panc-1cells were arrested at the G2/M phase after CP treatment (Figure 4A). Compared to the 15.82% ± 2.05% G2/M phase in the control group, cells in the G2/M phase rose to 28.42% ± 1.42% in cells treated with a CP concentration of 50 μg/mL.

Furthermore, we examined the expression of proteins related to cell cycle. Checkpoint kinase 1 phosphorylation (Chk1) is essential for DNA damage-induced cell cycle arrest, resulting in G2 phase blockage [41]. In our study, the expression level of p-chk1 was downregulated with a decline of p-cdc2 and cyclin B1 after 48 h CP treatment. Meanwhile, the expression of p21 was increased, which indicated that CP can interfere with cell cycle checkpoints to cause G2/M arrest (Figure 4B).

### 2.5. CP Suppressed Panc-1 Migration In Vitro by Regulating EMT

Most advanced-stage patients of pancreatic cancer were characteristic as high metastasis, and those who have relapsed usually found liver metastasis [42]. We performed wound-healing experiment to investigate whether CP has an effect on the migration of Panc-1 cells. As shown in Figure 5A, CP significantly inhibited cell migration in Panc-1 cells at a concentration of 6.5 μg/mL, which is a non-fatal concentration.

Evidence has shown that the epithelial–mesenchymal transition (EMT) contributes to metastasis by enhancing tumor migration and progression [43]. Thus, we examined the expression of EMT-related proteins. The decline mesenchymal cell markers, including Snail, Vimentin, and N-cadherin were observed (Figure 5B). We also found that CP restored E-cadherin expression after 48 h treatment, indicating that CP inhibited migration of Panc-1 cells by reversing the EMT process.

### 2.6. CP Exerted Anti-Tumor Effect on Panc-1 Cells via Hippo-YAP Pathway

Flavonoids, the most abundant component in propolis, were reported to inhibit the proliferation of cancer cells by activating the Hippo-YAP pathway [44,45], and Hippo-YAP is critical for pancreatic cancer development. Combining with our research, we inferred that the Hippo-YAP pathway might also play a role in the anticancer effect of CP. A previous study observed an increase in the nuclear location of YAP in pancreatic cancer [21]; thus, we examined YAP nuclear accumulation by immunofluorescence staining. The results (Figure 6A) showed nuclear-localized YAP was decreased after CP treatment for 48 h.

Additionally, we examined the expression of main components in the Hippo pathway (Figure 6B). Up-regulation of the upstream proteins LAST1 and MST1 were observed, with the activation of downstream effector protein YAP, confirmed with the increased expression of phosphorylated YAP (p-YAP) (Figure 6B). This result suggested the Hippo-YAP pathway was activated after CP treatment.

## 3. Discussion

Despite advances in diagnostic tools and treatment methods for pancreatic cancer, the incidence of pancreatic cancer is still continually rising, which was estimated to have reached 484,486 in 2020 [2,8]. Growing research is paying attention to the anti-tumor effect of CP, and previous studies have shown that CP can inhibit the growth of different cancers [33,46]. However, there are few studies addressing the effect of CP in pancreatic cancer. In this study, we firstly revealed the inhibitory effect of CP on pancreatic cancer cells in vitro and investigated the mechanism under it, which could not only bring new methods to treat pancreatic cancer but also lay the groundwork for further application of propolis.

Since the biological activities of propolis are determined by its components, we identified the chemical composition of the CP we used in this study. The results showed that chrysin, pinocembrin, 3-O-Acetylpinobanksin, galgangin, and CAPE were the main components (Figure 1), which was consistent with a previous study [47]. Mexico propolis and its components, including chrysin, pinocembrin, and 3-O-acetylpinobanksin, has been shown to have a cytotoxic effect on Panc-1 cells [48]. To evaluate whether Chinese propolis has an inhibitory effect on pancreatic cancer cell viability, we conducted a CCK-8 assay. In this study, CP showed a time- and dose-dependent inhibitory effect on Panc-1 cell viability (Figure 2). Consistent with the decreased cell viability, the cell apoptosis rate significantly rose as CP concentration increased (Figure 3A). To investigate the molecular mechanism behind this, we examined the expression of apoptosis-related proteins by western blot. Activation of caspase cascade was observed by increasing cleaved caspase initiator (caspase-8 and caspase-9) and executioner (caspase-3 and PARP). Apoptosis is a complex biochemical mechanism, and the extrinsic (death receptor pathway) and intrinsic (mitochondrial pathway) are considered the two main apoptotic pathways [49]. The intrinsic pathway of apoptosis is dependent upon the BCL-2 (B-cell CLL/Lymphoma 2) family for the efficient release of pro-apoptotic factors. We examined the expression of proteins from the Bcl-2 family and observed an increase of pro-apoptotic protein Bax, while anti-apoptotic protein Bcl-2 declined after 48h CP treatment. Our results demonstrated that intrinsic apoptosis was triggered in CP-treated Panc-1 cells (Figure 3B). Additionally, we found that Panc-1 cells were arrested in the G2/M phase after CP treatment (Figure 4A). G2/M phase was regulated by the complex formed by Cyclin B1 and cdc2 (also known as CDK1) [50]. In order to study the specific mechanism of CP influencing the cell cycle, we detected the expression of proteins related to cell cycle checkpoints by Western blot. After CP treatment, the expression of p-chk1 was down-regulated (Figure 4B), which led to the suppression of downstream factors p-cdc2 and cyclin B1 [51]. The down-regulation of p-cdc2, along with the decline of Cyclin B1, contributed to DNA-damage-induced G2/M arrest [52]. Besides, P21, as a CDK inhibitor, can inhibit G2/M progression by directly interacting with the CyclinB-cdc2 complex [41,53], which was also decreased in the current study.

The high mortality rate of pancreatic cancer is partly due to its metastatic characteristics [42,54]. The metastatic rate of Panc-1 cells was decreased significantly after CP treatment (Figure 5A), revealed by a wound healing test. EMT is regarded as a critical step of metastasis. Cells dissociate from the epithelial layer, deregulates cell-cell junctions, and enhances migratory capacity [55]. The lack of E-cadherin is considered a mark of EMT [56]. A previous study has shown that repression of E-cadherin could promote a metastatic phenotype in many cancer types including pancreatic cancer [57]. Along with the reduction of E-cadherin, an increase of mesenchymal marker proteins including N-cadherin, vimentin, and smooth muscle actin were also observed during the EMT process [58]. Snail is one of the EMT-inducing transcription factors, which can repress E-cadherin expression by interacting with chromatin remodeling factors [59,60,61]. Studies have reported that propolis and its components can regulate metastasis of cancer cells by inhibiting EMT [62,63,64,65,66]. Thus, we performed western blot to examine whether CP had an effect on the EMT process of Panc-1 cells. In this study, Snail and mesenchymal proteins such as N-cadherin, Vimentin were decreased, while E-cadherin was increased after CP treatment (Figure 5B), which indicated that CP inhibited pancreatic cancer metastasis by inhibiting the EMT.

The Hippo signaling pathway is associated with organ size control, cell differentiation, and metabolism regulation [67,68]. The Hippo pathway fulfilled its function by inactivating YAP actives. When the Hippo-YAP pathway is activated, the upstream kinases MST1/2 phosphorylate LATS1/2 by binding with the adapter protein SAV1/WW5. Then activated LAST1/2 phosphorylate and inactivate the effector protein YAP/TAZ by interacting with MOB1. The Phosphorylated YAP/TAZ then sequestered into the cytoplasm and eventually being degraded [69]. When the pathway was inactivated, it will lead to the overexpression and dephosphorylation of YAP/TAZ. Overexpressed YAP was translocated into nuclear to achieve its oncogenic function by interacting with transcriptional factors, including TEADs [70]. Pancreatic cancer was reported to express high-level YAP, which is confirmed to promote tumor development [71]. In our study, CP inhibited the nuclear localization of YAP (Figure 6A) and suppressed its expression in pancreatic cells. After CP treatment, expression of LATS1, MST1, and p-YAP was upregulated (Figure 6B), which suggested the activation of the Hippo pathway. These results indicated that the Hippo-YAP signaling pathway has a role in the anti-tumor effect of CP in Panc-1 cells. Furthermore, the Hippo pathway has been implied to interact with multiple pathways including mTOR [72], Wnt [73], Hedgehog [74], and Notch [75], to regulate the proliferation of cancer cells. Song et al. found YAP could inhibit apoptosis by binding with TEAD1 to regulate Bcl-2 activity [76]. Another study indicated that the Hippo pathway takes part in EMT by promoting the expression of FOXC2, Snail, and Twist [77]. Besides, YAP can promote EMT by binding with transcriptional factor FOS to upregulate the expression of Slug and Vimentin [78]. In our study, we also observed the down-regulation of Snail and Vimentin along with the decline of YAP after CP treatment. Additionally, LATS1 activation leads to cell cycle arrest at the G2/M phase by regulating Cyclin kinase/cdc2 activities [79]. Combined with our results, the Hippo pathway may have a critical role in regulating Panc-1 cells by integrating with several other pathways after CP treatment.

Our study was conducted in pancreatic cancer cells in vitro. In future studies, we should purify, characterize and synthesize the active components in CP as well as transfer in vitro cell-based experiments to in vivo test systems for further verification. Currently, our findings only provide limited and preliminary evidence. Further studies should be carried out to determine the main effective components in propolis that induce the anticancer effect.

## 4. Materials and Methods

### 4.1. Chemicals and Reagents

Vanillic, caffeic acid, rutin, ferulic acid, isoferulic acid, p-coumaric acid, cinnamic acid, 3,4-dimethoxycinnamic acid, caffeic acid phenethyl ester (CAPE), myricetin, apigenin, galangin, chrysin, pinocembrin, quercetin, kaempferol, luteolin, and naringenin were purchased from Sigma–Aldrich (St. Louis, MI, USA), pinobanksin and 3-O-acetylpinobanksin were purchased from Ningbo Haishu Apexocean Biochemicals Co., Ltd. (Ningbo, China). High glucose Dulbecco’s Modified Eagle’s Medium (DMEM) and fetal bovine serum (FBS) were purchased from Gibco (NY, USA). 5-Fluorouracil (5-FU) was purchased from Sigma (St. Louis, MI, USA). Propidium iodide (PI), 4′6-diamidino-2-phenylindole (DAPI), and dimethyl sulfoxide (DMSO) were purchased from Sangon Biotechnology. Co. Ltd. (Shanghai, China). Cck-8 kit and the Annexin V Fitc apoptosis detection kit were purchased from Dojindo (Kumamoto, Japan). Cell cycle and apoptosis analysis kits were purchased from Beyotime (Shanghai, China). Primary antibodies against pro-caspase-3, pro-caspase-8, pro-caspase-9, cleaved-caspase-3, cleaved-caspase-8, cleaved-caspase-9, PARP, cleaved-PARP, p-Chk1, p-cdc2, cyclin B1, Yap, p-Yap, MST1, tublin, anti-rabbit secondary antibodies, and goat anti-rabbit Alexa Fluor 594 secondary antibody were purchased from Abcam (Cambridge, MA, USA). Primary antibodies against Snail, E-cadherin, N-cadherin Vimentin, and LATS1 were purchased from Cell Signaling Technology (Danvers, MA, USA).

### 4.2. Preparation of CP

Raw CP was collected from the colony of *Apis mellifera* L. in Shandong Province, of which the main plant source was poplar. Raw propolis was extracted with 95% (*v/v*) ethanol at 40 °C for 4 h in an ultrasonic water base, then put in a 4 °C refrigerator overnight and filtered the mixture solution with a paper filter; the extraction was repeated three times. The solution is then evaporated in a rotary evaporator at reduced pressure at 40 °C until a constant weight is reached. The resulted solution was redissolved in ethanol (95%) to make a 50 mg/mL stock solution which was stored under a dry condition at −20 °C for further use.

### 4.3. Cell Culture

Panc-1 cells were purchased from the Cell Bank of the Chinese Academy of Sciences (Shanghai, China). Hek293 cells were gifted by Zhejiang University of Traditional Chinese Medicine. Both Panc-1 cells and Hek293 cells were cultured in DMEM medium supplemented with 10% FBS and 100 U/mol of penicillin at 37 °C in a humidified 95–5% (*v*/*v*) air and CO_2_, respectively. Cells were grown to 80–90% confluence before drug administration.

### 4.4. Identification of CP by HPLC

Major compounds in CP were determined by HPLC as described in a previous study [18]. The separation was conducted on a Sepax HP-C18 column (150 mm × 4.6 mm, 5 μm; Sepax Technologies, Inc., Newark, DE, USA), the flow rate is 1.0 mL/min at 30 °C, and the injection volume was 5 μL. The mobile phases used were 0.1% aqueous acetic acid (*v*/*v*; mobile phase A) and methanol (mobile phase B), and the linear gradient model were set as follows: 25% to 55% (B) at 0 to 30 min, 55% to 80% (B) at 30 to 60 min, 80% to 95% (B) at 60 to 70 min, and 95% to 25% (B) at 70 to 90 min. The UV detector was performed at 280 nm.

### 4.5. Cell Viability Assay

The cell viability was assessed by using a CCK-8 assay following the manufacturer’s instructions. Panc-1 cells and Hek293 cells were seeded at the density of 2 × 10^4^/mL into 96-well plates for 24 h, then treated with varying concentrations of CP (6.25, 12.5, 25, 50, 100, and 200 μg/mL). After 24 h and 48 h treatment, 10 μL of CCK-8 solution was added to cells and were then incubated at 37 °C for 2 h. The optical density (OD) was measured by using a microplate reader (Bio-Rad Model550, CA, USA) at 450 nm.

### 4.6. Cell Apoptosis Assay

The cell apoptosis was assessed by using Apoptosis Detection Kit following the manufacturer’s instructions. After 48 h treatment of CP (12.5, 25, and 50 μg/mL), Panc-1 cells were harvested and washed twice with phosphate-buffered saline, then centrifuged. Cells were seeded into 6-well plates and treated with Annexin V-FITC and PI binding solution, then analyzed by flow cytometry with CellQuest (NJ, USA).

### 4.7. Cell Cycle Assay

Panc-1 cells were cultured in 60 mm culture dishes at a density of 4 × 10^5^ at 37 °C overnight; then, the cells were treated with 12.5, 25, and 50 μg/mL of CP for 48 h. After treatment, cells were collected and treated following the protocol of the Cell Cycle and Apoptosis Analysis kit. The cell cycle distribution was analyzed by flow cytometry with CellQuest (NJ, USA).

### 4.8. Wound Healing Assay

Two-well culture inserts (ibidi GmbH, Planegg, Germany) were used for the wound healing assay in vitro following the manufacture’s protocol. After 24 h, the insert was removed, forming a 500 mm width cell-free gap. Then the cells were treated with 6.25 μg/mL CP, which is a non-fatal concentration. The migration rate was assessed by the healing areas, which were documented every 24 h with a microscope.

### 4.9. Immunofluorescence Staining

Panc-1 cells were seeded in 20 mm glass-bottom cell culture dishes (NEST, Wuxi, China) to appropriate density, then fixed in 4% paraformaldehyde for 10 min and permeabilized with 0.1% Triton X-100 for 10 min. After blocking with 3% BSA for 30 min, cells were incubated with the primary antibodies: monoclonal anti-YAP antibody diluted in 1% BSA in 4 °C overnight. Then cells were washed with PBS and incubated with Alex Flour 594-conjugated goat anti-rabbit IgG (1:250 dilution) for 1 h in the dark. The cells were then washed and stained with DAPI solution for 5 min. The immunofluorescence result was obtained with a confocal laser microscope (Leica TCS SP5, Wetzlar, Germany).

### 4.10. Western Blot

Panc-1 cells were seeded into 6-well plates at a density of 1 × 10^6^ cells/well with varying concentrations of CP. After 48 h treatment, cells were washed twice with pre-cold PBS twice on ice. Proteins were lysed with RIPA (Roche, Basel, Switzerland). Collecting and vortexed the cell lysate at 12,000 rpm for 10 min, then put on ice for 10 min to remove the cell debris. Mixed the cellular protein with sample loading buffer (Fudebio, Hangzhou, China), then boiled the mixture at 95 °C for 10 min. The BCA protein assay kit (Fudebio, Hangzhou, China) was used to measure the protein concentration, and 12% SDS-PAGE (Fudebio, Hangzhou, China) was used to separate the protein, which was then transferred to PVDF membranes (Millipore, Billerica, MA, USA). Blocked PVDF membranes with 5% skim milk (BD, NY, USA) dissolved in TBST solution for 1 h at room temperature, which were then incubated the immunoblots with specific primary antibodies overnight at 4 °C. After primary antibody binding, incubating the blots with AP-conjugated anti-rabbit secondary antibodies for 1 h at room temperature. Beta-tublin was used as the internal control, and the immunoblots were developed with the ECL method.

### 4.11. Statistical Analysis

At least three experiments were conducted in each assay. All data are expressed as the mean ± SD. The Student’s *t*-test was used to determine statistical differences. *p* < 0.05 was considered statistically significant difference (* *p* < 0.05; ** *p* < 0.01). Statistical analyses were performed using GraphPad Prism 6.0 (GraphPad Software Inc., CA, USA).

## 5. Conclusions

In conclusion, our study discovered an anti-tumor effect of CP on pancreatic cancer cells in vitro. In this study, CP induced apoptosis and promoted cell cycle arrest in Panc-1 cells, which were considered to be the main mechanisms for CP to inhibit pancreatic cancer viability. Moreover, in vitro migration assays showed that CP suppressed Panc-1 migration by inhibiting the EMT process. Furthermore, we found that the Hippo-YAP signaling pathway was activated, which could be the mechanism of the anticancer effect of CP. Our results could shed light on the potential pharmacological use of CP for pancreatic cancer treatment.

## Figures and Tables

**Figure 1 molecules-26-02803-f001:**
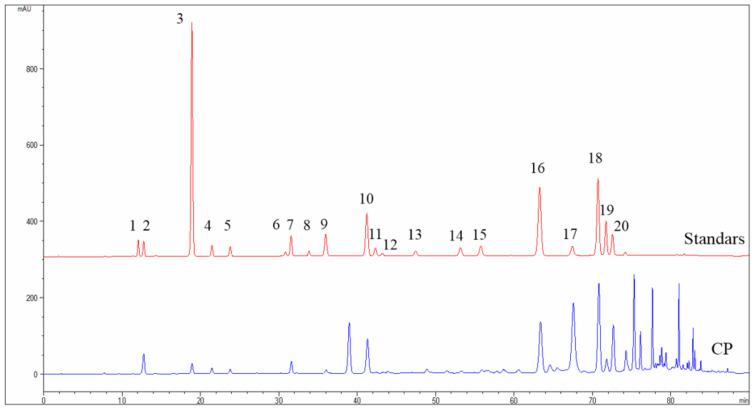
The HPLC chromatograms of standard solution and CP. 1: Vanillic; 2: Caffeic acid; 3: p-Coumaric acid; 4: Ferulic acid; 5: Isoferulic acid; 6: Rutin; 7: 3,4-Dimethoxycinnamic acid; 8: Myricetin; 9: Cinnamic acid; 10: Pinobanksin; 11: Naringenin; 12: Quercetin; 13: Luteolin; 14: Kaempferol; 15: Apigenin; 16: Pinocembrin; 17: 3-O-acetylpinobanksin; 18: Chrysin; 19: CAPE; 20: Galangin.

**Figure 2 molecules-26-02803-f002:**
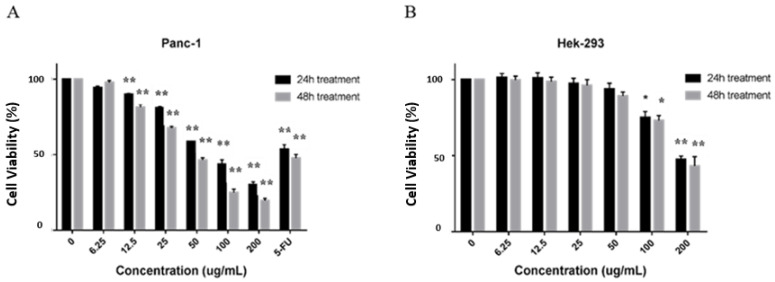
CP inhibited the viability of Panc-1 cells and had little effect on normal Hek-293 cells. (**A**) Panc-1 cells were treated with various concentrations of CP for 24 and 48 h. 50 μM Fluorouracil (5-FU) was used as a positive control. After treatment, CP exerted an anti-proliferation effect on Panc-1 cells. (**B**) CP showed no cytotoxic effect on normal HEK-293 cells until the concentration reaches 100 μg/mL. Data are presented as mean ± SD; * *p* < 0.05 and ** *p* < 0.01 versus control; *n* = 5.

**Figure 3 molecules-26-02803-f003:**
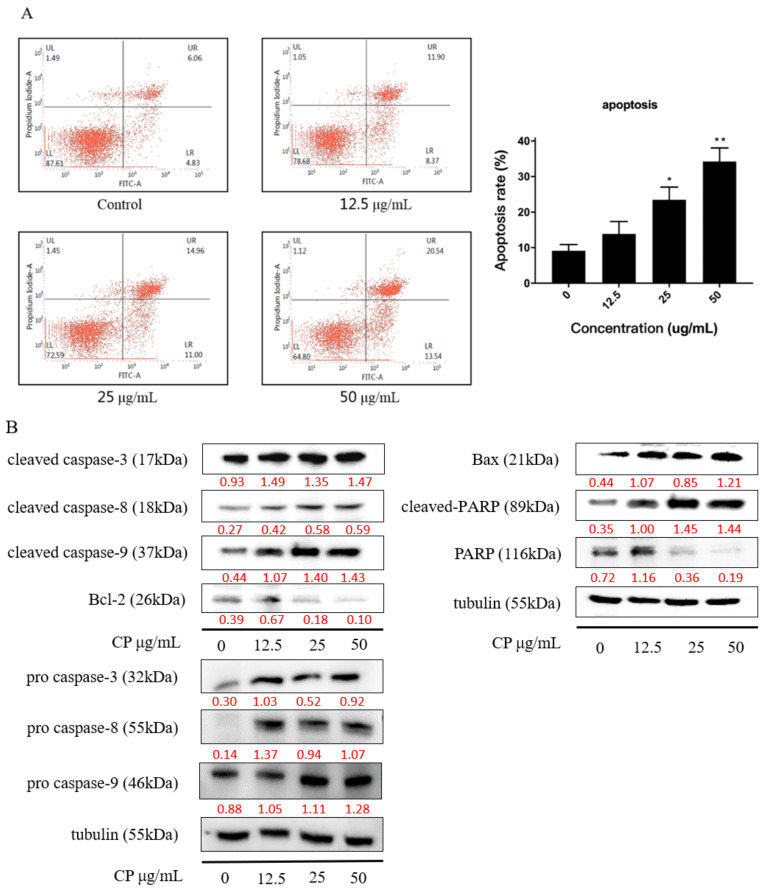
CP induced apoptosis in Panc-1 cells by activating a caspase cascade. (**A**) CP significantly increased the apoptosis rate of Panc-1 cells in a dose-dependent manner. (**B**) Caspase cascade was activated after CP treatment; the expression of the caspase family of proteins was analyzed by Western blot. Data are presented as mean ± SD; * *p* < 0.05 and ** *p* < 0.01 versus control; *n* = 3.

**Figure 4 molecules-26-02803-f004:**
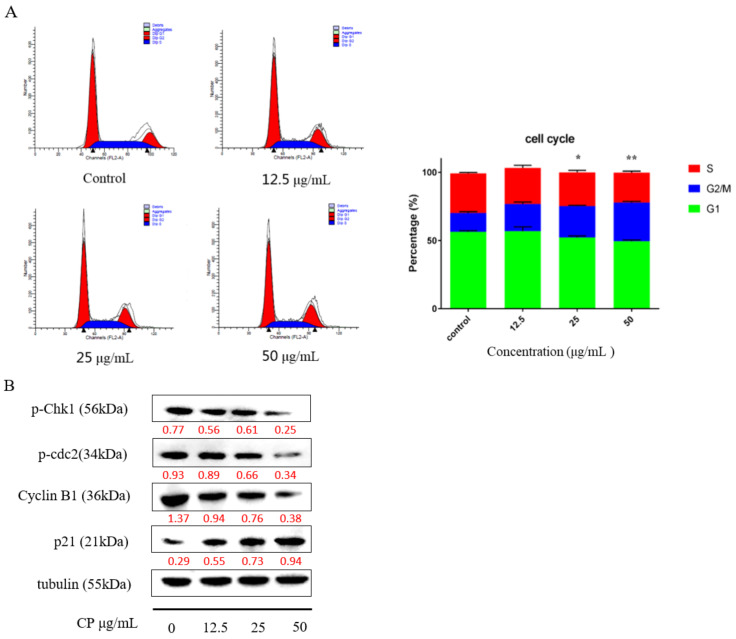
CP promoted G2/M phases cell cycle arrest in Panc-1 cells. (**A**) The cell cycle distribution after CP treatment at the concentration of 12.5, 25, and 50 μg/mL for 48 h. (**B**) Cell cycle checkpoint protein expression was analyzed by Western blot. Data are presented as mean ± SD; * *p* < 0.05 and ** *p* < 0.01 versus control; *n* = 3.

**Figure 5 molecules-26-02803-f005:**
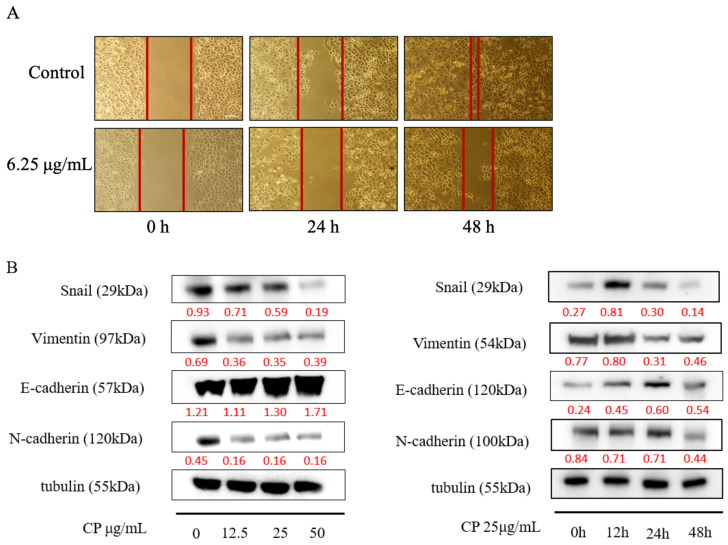
CP inhibited the migration of Panc-1 cells by inhibiting the EMT process. (**A**) CP suppressed migration of Panc-1 cells at a non-fatal concentration of 6.25 μg/mL. (**B**) The expression of EMT-related proteins was analyzed by Western blot with different concentrations and treatment times.

**Figure 6 molecules-26-02803-f006:**
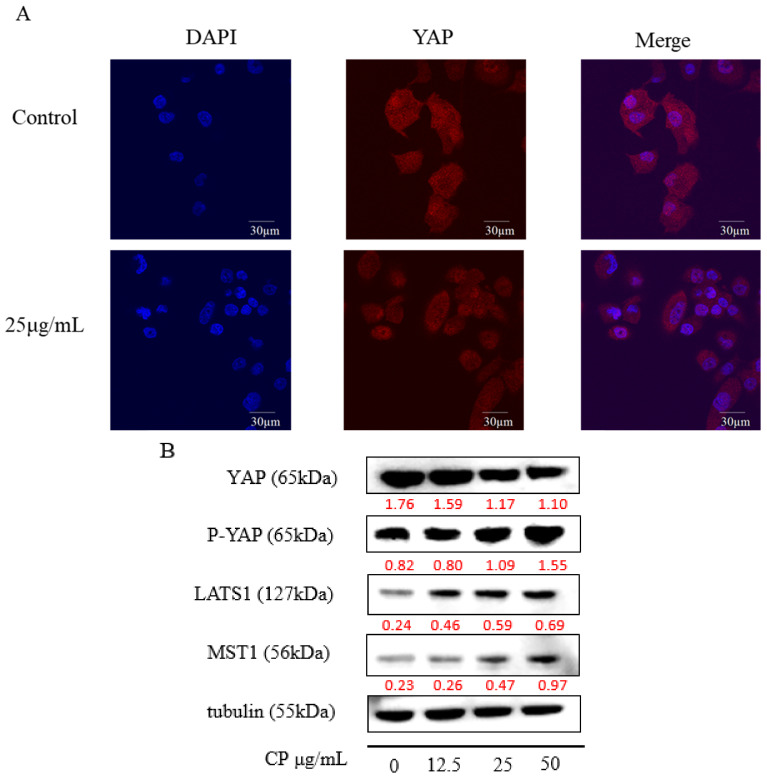
CP exerted an anti-tumor effect on Panc-1 cells by regulating the Hippo-YAP pathway. (**A**) Immunofluorescence image of YAP in Panc-1 cells treated with CP for 48 h. (**B**) The expression of the main components in the Hippo-YAP pathway, including YAP, p-YAP, LATS1, and MST1, detected by Western blot.

**Table 1 molecules-26-02803-t001:** The main components identified in CP by HPLC.

Peak Number	Compounds	Retention Time (min)	Contents (mg/g)
1	Vanillic	12.274	-
2	Caffeic acid	13.337	8.82
3	p-Coumaric acid	19.421	7.36
4	Ferulic acid	21.831	3.41
5	Isoferulic acid	24.079	5.07
6	Rutin	31.137	-
7	3,4-Dimethoxycinnamic acid	31.773	9.04
8	Myricetin	33.743	-
9	Cinnamic acid	32.597	0.87
10	Pinobanksin	41.856	30.21
11	Naringenin	42.603	-
12	Quercetin	42.894	1.45
13	Luteolin	47.682	-
14	Kaempferol	53.397	2.73
15	Apigenin	55.441	4.48
16	Pinocembrin	63.857	37.62
17	3-O-acetylpinobanksin	67.032	35.89
18	Chrysin	70.641	42.23
19	CAPE	71.865	15.66
20	Galangin	73.025	38.94

## Data Availability

The data presented in this study are available within the article.

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
