# Peer review of "Chinese Propolis Suppressed Pancreatic Cancer Panc-1 Cells Proliferation and Migration via Hippo-YAP Pathway"

_molecules, 2021, doi:10.3390/molecules26092803_

Round 1

Reviewer 1 Report

The authors have adequately addressed my initial review comments, which have improved the quality and accuracy of the manuscript.  English language/grammar would benefit by careful review.  Throughout the paper, the authors refer to the Panc-1 cells as pancreatic cancer; they should revise this to refer to them as pancreatic cancer cells, and clarify that the experiments are conducted in pancreatic cancer cells in vitro (such as on page 9, lines 209 and 211).  Furthermore, a discussion of the limitations of their experiments is suggested that includes the translational aspects of in vitro cell-based experiments to in vivo test systems, as well as identification and isolation of the active compound.

Author Response

Response to Reviewer 1 Comments

Point 1: Throughout the paper, the authors refer to the Panc-1 cells as pancreatic cancer; they should revise this to refer to them as pancreatic cancer cells, and clarify that the experiments are conducted in pancreatic cancer cells in vitro (such as on page 9, lines 209 and 211). 

Response 1: We sincerely appreciate the reviewer’s careful review of our manuscript. We have made the correction on page 9, lines 210 in blue.

Point 2: Furthermore, a discussion of the limitations of their experiments is suggested that includes the translational aspects of in vitro cell-based experiments to in vivo test systems, as well as identification and isolation of the active compound.

Response 2: We thank the reviewer’s suggestion. And we added a paragraph to discuss the limitations of our experiment on page 10, line 276-281 as following:

Our study was conducted in pancreatic cancer cells in vitro. If taking it all into consideration, we should purify, characterize and synthesize the active components in CP as well as transfer in vitro cell-based experiments to in vivo test systems for further verification. Now our findings only provided limited and preliminary evidence. Further study should be carried out to determine the main effective components in propolis that induce the anti-cancer effect.

Reviewer 2 Report

The study Tao et al. aims at showing the activity of CP on Panc-1 cells. The limitations of the report are (1) one cancer cell lines used; (2) poor quality of data (especially Western blots) with regard to support conclusions; (3) numeros typos.

Specific comments:

  1. Fig. 2: viality should be corrected to viability.
  2. Western blot images should be quantified, and molecular weight markers should be indicated.
  3. Fig. 3b: tublin should be changed to tubulin (similarly Fig. 4b and 5b and 6b); it would be more logical to pro-caspase and cleaved caspase directly below etc. In addition, results on caspase activation are very inconsistent. Why control cells have so much activated caspases (differences between control and treated cells are minor)? These results do not support the conclusion that intrinsic pathway of apoptosis is induced.
  4. Fig. 4: total Chk1 and cdc2 proteins should be also provided.
  5. Fig. 5: if CP was efficient at 6.25 ug/ml in inhibition of cell migration, why in 5B western blots were performed using substantially higher concentration?
  6. Fig. 6b: photographs are not convincing considering the conclusion. Similarly, presented western blots. 
  7. The general conclusion (e.g., in the title) is not supported by data. The Authors did not investigate directly the contribution of Hippo pathway and the activity of CP on cell migration. 
  8. Discussion is a repetition of result description.

Reviewer 3 Report

Molecules-11504229

Dear Authors,

I read the manuscript “Chinese propolis suppressed Pancreatic cancer Panc-1 cells proliferation and migration via Hippo-YAP pathway” very carefully and I think it can be accepted for publication in Molecules after major revision.

In the pdf version of the manuscript, many parts of the text are highlighted in yellow and there are also words in red. It appears that the authors have submitted a draft. Please check that the submitted version is complete.

The manuscript needs extensive grammar revision and correction of typography errors.

Here are some suggestions for improving the manuscript.

Abstract

Line 14: The number fifty of “IC50” must be written in the subscript. Check that in all the text "IC50" is written in subscript.

  1. Introduction

Write "etc." without inserting References is of little use. Please remove "etc." throughout the text or insert the necessary Reference.

Line 32: remove “etc.”

Lines 38-39: replace “resulting in a situation that most patients are diagnosed at advanced stage with high metastasis [10]” with “resulting in a situation that most patients are diagnosed at an advanced stage with high metastases [10], and diffuse malignant peritoneal mesothelioma [insert the reference with doi.org/10.1021/acs.jmedchem.6b00777].

Line 55: remove “and etc.”

Lines 73-74: replace “Chinese propolis (CP) is generally poplar type of which the main chemical components are flavonoids and phenolic compounds [28, 29].” with “Chinese propolis (CP) is generally of the poplar type whose main chemical components are flavonoids and phenolic compounds [28, 29, insert the reference with doi.org/10.1023/A:1021528628524].”

Line 76: remove “and etc.”

  1. Results

It would be useful to insert a figure with the chemical structures of the six most abundant components extracted from the CP.

There are very intense peaks in the chromatogram between 75-84 minutes. Why did the Authors not investigate to identify these compounds as well? Why did the Authors not perform a high resolution HPLC-MS analysis?

Line 89: in the title “2.1. Chemical composition of CP indentified by high-performance liquid chromatography (HPLC)”, replace “indentified” with “identified”

Line 95: replace “And” with “Furthermore”

Line 109: in the title “2.2. CP exerted cytotoxitic effect on Panc-1 cells”, replace “cytotoxitic effect” with “cytotoxic effect”

Line 123: in the title “2.3. CP induced apoptosis in Panc-1 cells via intrinsic way”, replace “via intrinsic way” with “via intrinsic pathway”

Line 140: replace “via intrinsic way” with “via intrinsic pathway”

Replace title “2.5. CP supressed migration of Panc-1 in vitro by regulating EMT” with “2.5. CP suppressed Panc-1 migration in vitro by regulating EMT”

Line 162: replace “had reoccurrence” with “have relapsed”

Lines 165, 168: replace “nonfatal concentration” with “non-fatal concentration”

Line 170: replace “treat time” with “treatment time”

  1. Discussion

Line 201: replace “pancreatic cancer in vitro” with “pancreatic cancer cells in vitro”

Lines 202-204: replace “which may not only bring new methods for pancreatic cancer treatment but also lay a foundation for further application of propolis.” with “which could not only bring new methods to treat pancreatic cancer, but also lay the groundwork for further application of propolis.”

Line 213: replace “viability. (Figure 2)” with “viability (Figure 2).”

Line 218: replace “mechanism, the extrinsic” with “mechanism, and the extrinsic”

Line 251: remove “and etc.”

  1. Materials and Methods

4.1. Chemicals and Reagents

Line 279: insert “caffeic acid phenethyl ester (CAPE)”

Line 288: replace “Cell cycle and apoptosis analysis Kit was purchased from” with “Cell cycle and apoptosis analysis Kits were purchased from”

4.2. Preparation of CP

Lines 301-302: replace “After that evaporating the solution in a rotary evaporator under a reduced pressure at 40°C until reaching a constant weight” with “The solution is then evaporated in a rotary evaporator at reduced pressure at 40°C until a constant weight is reached”

Line 344: replace “nonfatal concentration” with “non-fatal concentration”

  1. Conclusions

Lines 376-377: replace “pancreatic cancer in vitro” with “pancreatic cancer cells in vitro”

Line 377: replace “promote” with “promoted”

Line 378: replace “which consider to be” with “which are considered to be”

Lines 380-381: replace “Besides, we found that Hippo-YAP signaling pathway was activated which may be the mechanism of CP’s anti-tumor effect” with “Furthermore, we found that the Hippo-YAP signalling pathway was activated, which could be the mechanism of the anticancer effect of CP.”

Best regards.

Round 2

Reviewer 2 Report

All comments have been adressed in a satisfactory way.

Reviewer 3 Report

The authors have improved the manuscript with corrections, although the composition of Chinese propolis has not been fully clarified. Overall, the manuscript can be accepted in Molecules.